# Regulatory T Cells Exhibit Interleukin-33-Dependent Migratory Behavior during Skin Barrier Disruption

**DOI:** 10.3390/ijms22147443

**Published:** 2021-07-12

**Authors:** Sumika Toyama, Catharina Sagita Moniaga, Susumu Nakae, Masaru Kurosawa, Hideoki Ogawa, Mitsutoshi Tominaga, Kenji Takamori

**Affiliations:** 1Juntendo Itch Research Center (JIRC), Institute for Environmental and Gender Specific Medicine, Juntendo University Graduate School of Medicine, 2-1-1 Tomioka, Urayasu, Chiba 279-0021, Japan; su-toyama@juntendo.ac.jp (S.T.); m-catharina@juntendo.ac.jp (C.S.M.); mskurosa@juntendo.ac.jp (M.K.); ogawa@juntendo.ac.jp (H.O.); tominaga@juntendo.ac.jp (M.T.); 2Graduate School of Integrated Sciences for Life, Hiroshima University, 1-4-4 Kagamiyama, Higashi-Hiroshima, Hiroshima 739-8528, Japan; snakae@hiroshima-u.ac.jp; 3Department of Dermatology, Juntendo University Urayasu Hospital, 2-1-1 Tomioka, Urayasu, Chiba 279-0021, Japan

**Keywords:** acanthosis, dry skin, homeostasis, IL-33, skin barrier, Treg

## Abstract

Regulatory T cells (Tregs) suppress immune responses and maintain immunological self-tolerance and homeostasis. We currently investigated relationships between skin barrier condition and Treg behavior using skin barrier-disrupted mice. Skin barrier disruption was induced by repeated topical application of 4% sodium dodecyl sulfate (SDS) on mice. The number of CD4^+^ forkhead box protein P3 (Foxp3)^+^ Tregs was higher in 4% SDS-treated skins than in controls. This increasing was correlated with the degree of acanthosis. The numbers of interleukin (IL)-10^+^ and transforming growth factor (TGF)-β^+^ Tregs also increased in 4% SDS-treated skins. Localization of IL-33 in keratinocytes shifted from nucleus to cytoplasm after skin barrier disruption. Notably, IL-33 promoted the migration of Tregs in chemotaxis assay. The skin infiltration of Tregs was cancelled in IL-33 neutralizing antibody-treated mice and IL-33 knockout mice. Thus, keratinocyte-derived IL-33 may induce Treg migration into barrier-disrupted skin to control the phase transition between healthy and inflammatory conditions.

## 1. Introduction

The skin is an important barrier that protects against invasion by foreign substances, including harmful microorganisms and irritants, and retains water in the body [1,2]. Several dermatological conditions, including atopic dermatitis (AD), are associated with dry skin resulting from skin barrier disruption [3,4]. Indeed, downregulation of skin hydration factors such as natural moisturizing factors (NMFs) and molecules indispensable to skin barrier homeostasis, vinculin and late cornified envelope 2A are found in skins of AD patients [4].

Previous studies revealed that regulatory T cells (Tregs) are involved in the negative regulation of immune responses [5]. Naturally occurring CD25^+^CD4^+^ Tregs, which express the transcription factor *forkhead box protein P3* (Foxp3) [5,6,7,8], actively maintain immunological self-tolerance and homeostasis through the secretion of cytokines such as interleukin (IL)-10 and transforming growth factor (TGF)-β [9,10]. The fatal multiorgan autoinflammatory destruction in both immune dysregulation, polyendocrinopathy, enteropathy, and X-linked syndrome (IPEX) patients and scurfy mice is caused by mutations in the transcriptional regulator Foxp3 [11,12]. Natural mutations in the *FOXP3* gene cause the fatal autoimmune Scurfy phenotype in mice and IPEX syndrome in humans [7,13,14,15]. The most common clinical features of IPEX are severe secretory diarrhea, type 1 diabetes mellitus, and severe eczema; less common symptoms include thyroiditis, hemolytic anemia, thrombocytopenia, autoimmune hepatitis, and recurrent infections [14,15]. This raises the possibility the correlation between Tregs and skin homeostasis. However, the role of Tregs in skin barrier disruption remains unclear.

In this study, we investigated the relationship between the skin barrier condition and skin infiltration of Tregs in a mouse model of skin barrier disruption. Herein, we describe that cutaneous Tregs exhibit distinct migratory behavior during skin barrier damage.

## 2. Results

### 2.1. Establishment of Skin Barrier Disruption Model Mice by Repeated 4% SDS Application

We use 4% sodium dodecyl sulfate (SDS) application to induce skin barrier disruption in AD model NC/Nga mice, as described previously [16]. To disrupt the skin barrier, we topically applied 4% SDS to the dorsal skin of NC/Nga mice repeatedly (Scheme 1). Erythema and/or dryness were slight by 4% SDS application (Figure 1a), although the dermatitis score in 4% SDS-treated mice on days 11 to 18 was significantly higher than that in naïve or vehicle-treated mice (Figure 1b). In addition, transepidermal water loss (TEWL) values in 4% SDS-treated mice increased gradually over time and they were higher than those in naïve or vehicle-treated mice, confirming the skin barrier disruption (Figure 1c). No significant differences in scratching behavior or serum total IgE levels were found when comparing naïve, vehicle-treated, and 4% SDS-treated mice at any time during the induction and recovery phases. Representative data for the third week are shown in Figure 1d,e.

### 2.2. Effects of Skin Barrier Disruption on Acanthosis and Inflammatory Cells in Lesional Skin

We next characterized skin barrier disruption model mice. On hematoxylin-eosin (HE) staining, acanthosis was observed in 4% SDS-treated mice (Figure 2a). Effects of skin barrier disruption on the number of cutaneous inflammatory cells were examined by histological and FACS analysis. In comparison with control mice, repeated application of 4% SDS for 3 weeks did not affect the numbers of cutaneous mast cells, eosinophils, and neutrophils (Figure 2b–d). In contrast, the number of CD4^+^ T cells in the dermis significantly increased in 4% SDS-treated mice (Figure 2e). Alteration of the number of Tregs in affected skin was also demonstrated by in situ hybridization. Most CD4^+^ T cells (98.2 ± 2.53%) were positive for Foxp3 (data not shown). The number of CD4^+^Foxp3^+^ Tregs in the dermis was significantly higher in 4% SDS-treated mice than in naïve and vehicle-treated mice (Figure 2f). Similar results were obtained by FACS analyses (Appendix A). The number of Tregs correlated positively with acanthosis (r = 0.6957, *p* = 0.004) (Figure 2g), whereas there were no correlations with the numbers of mast cells, eosinophils, and neutrophils (Appendix A).

### 2.3. Tregs Express Regulatory Cytokines in Barrier-Disrupted Skin

The effects of treatment with 4% SDS and its discontinuation (recovery phase) on epidermal thickness and the number of CD4^+^Foxp3^+^ Tregs were examined weekly (Figure 3a–d). Acanthosis was induced by repeated application of 4% SDS during the induction phase, whereas it gradually disappeared during the recovery phase (Figure 3a,b). CD4^+^Foxp3^+^ Tregs increased during the application of 4% SDS and then decreased 1 week after discontinuing treatment (Figure 3c,d), a finding not observed in naïve and vehicle-treated mice (Figure 3a–d). We further examined whether subtypes of Tregs produce regulatory cytokines, such as IL-10 and TGF-β, in the skin. In situ hybridization analyses revealed that the numbers of IL-10^+^Foxp3^+^ Tregs and TGF-β^+^Foxp3^+^ Tregs peaked at 3 weeks and were higher in 4% SDS-treated mice than in naïve and vehicle-treated mice. The numbers of these Tregs returned to normal levels at 5 weeks after discontinuation of 4% SDS treatment (Figure 3e–h).

### 2.4. Localization of IL-33 in Barrier-Disrupted Skin

Epithelial cell-derived IL-33 was previously suggested to regulate Treg migration in the skin, but there is no direct evidence [17]. Therefore, we focused on IL-33 as a factor that causes Treg migration to lesional skin. The distribution of IL-33 was examined immunohistochemically in skins of naïve, vehicle- or 4% SDS-treated mice. IL-33 exhibited only intranuclear localization in epidermal keratinocytes in the naïve and vehicle-treated mice (Figure 4a,b). In contrast, the staining pattern became cytoplasmic in the 4% SDS-treated mice (Figure 4c,d).

### 2.5. IL-33-Dependent Migratory Behavior of Tregs

We next examined whether IL-33 directly induces Treg migration. Tregs that infiltrated the skin and Tregs that isolated from the spleen both expressed the IL-33 receptor ST2 (Figure 5a, Appendix A). In vitro chemotaxis assay demonstrated that IL-33 significantly and dose-dependently induced CD4^+^CD25^+^ Treg migration more than the previously reported migration factors CC *chemokine* ligand (CCL) 17, CCL21, and CCL22 (Figure 5b). We further examined the effects of IL-33 neutralizing antibody on skin infiltration of Tregs at 3 weeks after 4% SDS treatment by in situ hybridization. CD4^+^ Foxp3^+^ Tregs increased in the control IgG-treated group, whereas they significantly decreased in the IL-33 neutralizing antibody-treated group (Figure 6a). This inhibition was also examined in skin barrier-disrupted *IL33*^−/−^ mice (Scheme 2). Tregs infiltrated the skin of *IL-33*^+/+^ mice after skin barrier disruption, whereas they did not infiltrate 4% SDS-treated skin of *IL-33*^−/−^ mice (Figure 6b).

## 3. Discussion

In the present study, we used a mouse model of skin barrier disruption, which was induced by the repeated application of the surfactant SDS. In this mouse model, repeated application of 4% SDS increased TEWL values, but the skin barrier condition normalized 2 weeks after discontinuation of 4% SDS application (Figure 1c). Our model also exhibited mild erythema and/or dryness 2 weeks after 4% SDS treatment, but it subsequently normalized without exacerbation of dermatitis (Figure 1a,b). This was consistent with the absence of itching and lack of increased serum IgE levels (Figure 1d,e). Of note, migratory behavior of Tregs was found in the 4% SDS-treated skin (Figure 2f, Appendix A). Our experimental environment consisted of a clean room, with almost no other antigens being detectable. Therefore, even if the skin barrier is broken by a surfactant, severe dermatitis such as atopic dermatitis does not occur due to the absence of environmental allergens such as mite antigens. Thus, our skin barrier-disrupted mouse model is useful for assessing Treg dynamics during skin barrier disruption. 

Naïve Tregs circulate in secondary lymphoid tissues such as lymph nodes, the spleen, and mucosa-associated lymphoid tissue (MALT), but they differentiate into effector Tregs (eTregs) by antigen stimulation and suppresses the inflammatory response [18]. Recently, eTregs were reported to not only suppresses the inflammatory response, but also to promote the repair response to wound healing. This suggests that eTregs function in the maintenance of tissue homeostasis [19]. Moreover, skin infiltration of CD4^+^CD25^+^Foxp3^+^ Tregs is impaired in lesional skin of AD patients and they play an immunosuppressive role in atopic diseases [20]. Therefore, these findings suggest that skin barrier disruption induces Treg migration to maintain skin homeostasis through the suppression of inflammation. 

Our histological data demonstrated that the application of 4% SDS for 3 weeks increased the numbers of IL-10^+^ and TGF-β^+^ Tregs (Figure 3e–h), whereas the numbers of eosinophils, neutrophils, and mast cells remained unchanged (Figure 2b–d). Tregs prevent allergic inflammation by inhibiting the action of mast cells, basophils, neutrophils, and eosinophils, and play an important role in tissue remodeling by suppressing resident tissue cells [21]. Tregs may also play a role in maintaining immune tolerance in healthy skin [22]. Thus, the lack of induction of the inflammatory events in our model may have been due to the increase in IL-10^+^ and/or TGF-β^+^ Tregs caused by skin barrier disruption. This suggests a partial role in controlling the phase transition between healthy and inflammatory conditions. Indeed, IL-10 mainly suppresses ILC2 and regulates innate immune systems, whereas TGF-β regulates adaptive immune systems [23,24,25]. In this study, our model mice had a long period of 3 weeks during which the adaptive immune system was active. Furthermore, the number of Tregs correlated with the degree of acanthosis (Figure 2g), which developed before Treg skin infiltration (Figure 3b,d). In the recovery phase, the increased number of Tregs normalized after the epidermal thickness normalized (Figure 3b). In addition, the numbers of IL-10^+^ Tregs and TGF-β^+^ Tregs peaked at 3 weeks, consistent with the results of CD4^+^Foxp3^+^ Treg behavior shown in Figure 3d. Although this study did not directly reveal that IL-10 and TGF-β were released simultaneously by following breakage of the skin barrier, these findings raise the possibility that IL-10 and TGF-β were functionally released at the same time (Figure 3e–h). Taken together, Tregs may involve in suppression of both the innate and adaptive immune systems in the skin barrier disruption model mice. 

Epithelial cell-derived chemokines, such as CCL17, CCL21, and CCL22, were reported to regulate Treg migration in the skin [26,27,28]. Moreover, a recent study revealed that the presence of Tregs in skin after repeated tape stripping is related to the increased expression of IL-33. In the model, IL-33 induced Treg development by shifting the dendric cells to a regulatory phenotype [17]. However, it has not been clarified how Treg are infiltrated in the skin and whether IL-33 is directly involved in their migration. In this study, our *in vitro* chemotaxis assay showed that IL-33 induced Treg migration more strongly than the previously reported chemokines CCL17, CCL21, and CCL22 (Figure 5b). Expressions of the CCL17, CCL21, and CCL22 are upregulated by IL-4 and IL-13, which accelerates the recruitment of Th2 cells and eosinophils during inflammatory responses [29,30]. Thus, although the responsiveness of Tregs to these chemokines or cytokines may differ in the presence of inflammation, our findings raise the possibility that IL-33 directly induces Treg migration into barrier-disrupted skin without the involvement of dendric cells. 

One limitation of this study was our use of isolated CD4^+^ CD25^+^ Tregs derived from spleen. Because Foxp3 is a transcription factor, positive staining requires cell fixation and permeabilization. To our knowledge, Foxp3 cannot be used as a sorting marker for the isolation of Tregs. FACS analysis of Foxp3 expression by isolated CD4^+^CD25^+^ cells showed that this factor was expressed in 29.2% of isolated cells (Appendix A). These findings suggested that one-third of the isolated CD4^+^CD25^+^ cells are Foxp3^+^ Tregs. 

IL-33 belongs to the IL-1 family, which mediates its biological effects through IL-33 receptor ST2 and accelerates the production of type 2 inflammation cytokines by in vitro polarized Th2 cells [31]. Our FACS analyses showed that the ST2 was expressed in Foxp3^+^ Tregs derived from skin and spleen (Figure 5a, Appendix A). Recent studies reported a ST2-expressing Treg subset called tisTregST2, which can be identified at high frequencies in peripheral tissues [32]. Our immunohistochemical analyses demonstrated that IL-33 proteins exhibit mainly intranuclear localization in epidermal keratinocytes in naïve mice, but it becomes cytoplasmic in treated skin (Figure 4). This alteration may imply an increase in extracellular secretion of IL-33. This idea is also supported by our *in vivo* experiments using anti-IL-33 neutralizing antibody or *IL-33*^−/−^ mice (Figure 5c,d). Therefore, epidermal keratinocyte-derived IL-33 may induce Treg infiltration directly associated with skin barrier disruption.

In conclusion, the present study demonstrated that skin infiltration of Tregs for recovery, especially those producing IL-10 and/or TGF-β, is induced by keratinocyte-derived IL-33 under skin barrier disruption. IL-33-dependent Treg migration may be partly responsible for controlling the phase transition between healthy and inflammatory conditions in healthy immune responses.

## 4. Materials and Methods

### 4.1. Animals

Male NC/Nga mice aged 5–9 weeks were purchased from Oriental Yeast (Tokyo, Japan). Male *Il33*^−/−^ mice and C57BL/6NCrl mice aged 8–9 weeks with an *Il33*^−/−^ background were purchased from Japan SLC, Inc. (Shizuoka, Japan). The genotype of *Il33*^−/−^ is shown in Appendix A. All mice were maintained in the experimental animal facility of Juntendo University Graduate School of Medicine under a 12-h light:12-h dark cycle at a regulated temperature of 22–24 °C, with food and tap water provided ad libitum. The study protocol was approved by the Institutional Animal Care and Use Committee at Juntendo University Graduate School of Medicine.

### 4.2. Skin Barrier-Disrupted Model Mice

Skin barrier disruption was induced in NC/Nga, C57BL/6NCrl, or *Il33*^−/−^ mice as described previously [33]. On the first day, hair was removed with an electric shaver and barrier disruption was induced by treatment of the shaved dorsal skin with 150 μL of 4% SDS twice weekly for three weeks (induction phase), followed by its discontinuation (recovery phase); control mice were treated with 150 μL deionized distilled water (DDW) (Scheme 1). 

To neutralize IL-33, 5 μg/200 μL of mouse IL-33 monoclonal antibody (R&D systems; Minneapolis, MN, USA) was intraperitoneally injected for 2 consecutive days prior to skin barrier disruption (Scheme 2).

### 4.3. Evaluation of Skin Condition and Dermatitis

Before and 2 h after each treatment, TEWL in the treated area was measured using a Tewameter^®^ TM210 (Courage & Khazawa, Cologne, Germany), as previously described [16]. To confirm the lesional skin condition, skin inflammation and itching were analyzed using dermatitis scoring and scratching behavior recording, respectively. Briefly, dermatitis scores were evaluated as previously described [16]. The severity of AD was assessed according to four symptoms: erythema/hemorrhage, scarring/dryness, edema, and excoriation/erosion. Each symptom was graded from 0 to 3 (none, 0; mild, 1; moderate, 2; severe, 3). The clinical skin score was defined as the sum of the individual scores and ranged from 0 to 12.

### 4.4. Observation of Scratching Behavior

Itch-related scratching behavior was analyzed as previously described with slight modification [16]. After the last 4% SDS application, mice (4 animals per observation) were placed in an acrylic cage (19.5 × 24 × 35 cm) for at least 1 h for acclimation. Scratching behavior was monitored for 12 h using a SCLABA^®^-Real system with observers out of the experimental room (Noveltec, Kobe, Japan).

### 4.5. Measurement of Total IgE Levels in Sera

Serum samples were collected weekly from each mouse. The total IgE level in each sample was measured using the LBIS Mouse IgE ELISA kit (FUJIFILM Wako Shibayagi Corp., Gunma, Japan) according to the manufacturer’s instructions.

### 4.6. Histological and Immunohistochemical Analyses

Murine skins were collected from the dorsal neck under sevoflurane anesthesia. Samples were fixed overnight at room temperature (RT) in 10% formalin neutral buffer (Wako Corp., Osaka, Japan), embedded in paraffin, and then sliced at a thickness of 4 μm. The sections were deparaffinized in xylene and dehydrated in an ethanol series. The samples were stained using the EMCSK (Diagnostic BioSystems, Pleasanton, CA, USA), May-Gruenwald (Wako Pure Chemical Industries Ltd., Osaka, Japan)–Giemsa (Muto Pure Chemicals Co., Ltd., Tokyo, Japan) solution, TB (Wako Pure Chemical Industries Ltd.), and HE solution (Wako Pure Chemical Industries Ltd.). 

For immunohistochemistry, skin sections were incubated in Antigen Retrieval Buffer (100× citrate buffer, pH 6.0) (abcam, Cambridge, UK) maintained at 98 to 100 °C using a microwave for 20 min. After the slides cooled to RT, they were blocked by incubation in PBS containing 2% bovine serum albumin (Sigma-Aldrich, St. Louis, MO, USA); 5% normal donkey serum (Merck Millipore Corp., Darmstadt, Germany) containing 0.2% Triton X-100 for 1 h at RT. The sections were incubated with rabbit anti-IL-33 antibody (1:50 dilution; ProSci, Poway, CA, USA) overnight at 4 °C, washed with PBS containing 0.05% Tween 20, and then incubated with the secondary antibody conjugated with Alexa Fluor dye (1:300 dilution; Thermo Fisher Scientific, Waltham, MA, USA) for 1 h at RT. The sections were cover-slipped with Vectashield-mounting medium for fluorescence with 4′,6-diamidino-2-phenylindole (DAPI) (Vector Laboratories, Inc., Burlingame, CA, USA). The slides were viewed under a BZ-X800 microscope (Keyence Corp., Osaka, Japan).

### 4.7. In Situ Hybridization Using the RNAscope^®^ Method

Skin samples were fixed overnight at RT in 10% formalin neutral buffer, embedded in paraffin, and then sliced at a thickness of 4 μm. Briefly, paraffin sections were deparaffinized in xylene and dehydrated in an ethanol series. The sections were incubated in RNAscope^®^ Target Retrieval Reagent (Advanced Cell Diagnostics, Newark, CA, USA) maintained at 98 to 100 °C using a hot plate for 15 min. The sections were rinsed in deionized water and then treated with RNAscope^®^ Protease III (Advanced Cell Diagnostics) at 40 °C for 30 min in a HybEZ™ Catalog Number hybridization was performed using the RNAscope^®^ Fluorescent Multiplex Reagent kit (Advanced Cell Diagnostics) and RNA probes (Advanced Cell Diagnostics) (Table 1) respectively, according to the manufacturer’s instructions [34].

### 4.8. Quantitative Analyses from Histological Data

Three specimens per mouse from 6–8 mice per group were stained with HE, TB, May-Gruenwald–Giemsa solutions, as well as with antibodies and RNA probes.

Epidermal thickness was measured using the BZ-X800 analyzer, at three randomly selected locations per HE stained section; and averaged for at least 36 bright field images per group in each experiment. The numbers of cutaneous mast cells, eosinophils, and neutrophils were hand-counted in at least 36 bright field images per group in each experiment. 

The number of epidermal keratinocytes positive for IL-33 were hand-counted in at least 36 fluorescence images were analyzed per group in each experiment. The number of CD4^+^ T cells, CD4^+^Foxp3^+^, TGF-β^+^Foxp3^+^, IL-10^+^Foxp3^+^ Tregs, and ST2^+^Foxp3^+^ Tregs were also counted in at least 36 fluorescence images per group in each experiment. The number of CD4^+^ T cells and each type Treg per 10^4^ μm was measured by the BZ-X800 analyzer.

### 4.9. FACS Analysis

Mice were anesthetized with sevoflurane, and single cell suspensions were obtained from the skin of dorsal neck by treatment with 400 U/mL collagenase type II (Worthington Biochemical, Freehold, NJ, USA), 1000 U/mL hyaluronidase Type 4S (Sigma Aldrich) and 50 U/mL DNase I (Roche; Basel, Switzerland) in HBSS (Nacalai Tesque, Kyoto, Japan) for 1 h. Samples were passed through a 70-μm pore size nylon mesh. The cells were pretreated with anti-CD16/32 (2.4G2 clone) in FACS Buffer (HBSS containing 2% FCS) for 20 min on ice. The cells were then stained (15 min on ice) with a mixture of PE-conjugated CD25 mAb (3C7; BioLegend; San Diego, CA, USA) and APC-conjugated CD4 mAb (GK1.5; BioLegend) or PE-conjugated Siglec-F mAb(E50-2440; BD; Franklin Lakes, NJ, USA), FITC-conjugated c-kit mAb (2B8; Biolenged), and Alexa Fluor^®^ 647-conjugated FcεRI (MAR-1; made from cell line in our laboratory). For intracellular staining, the cells were fixed and permeabilized using a Foxp3/Transcription Factor Staining Buffer Set (eBioscience; San Diego, CA, USA) and subsequently stained with Alexa Fluor^®^ 488-conjugated Foxp3 (MF-14; BioLegend), according to the manufacturer’s instructions. To detection of ST2^+^ and Foxp3^+^cells, spleen cells were suspended in HBSS containing 2% FCS passed through a 70-μm pore size nylon mesh, and stained with FITC-conjugated ST2 mAb (DJ8; MD Biosciences; Oakdale, MN, USA) or Alexa Fluor^®^ 488-conjugated Foxp3. Samples were acquired on a Facscalibur (BD Biosciences) and analyzed with FlowJo software (TreeStar, San Francisco, CA, USA).

### 4.10. In Vitro Chemotaxis Assay

Tregs were prepared from murine spleen samples using the CD4^+^CD25^+^ Regulatory T cell Isolation kit (Miltenyi Biotec, Bergisch Gladbach, Germany). The transwell apparatus (Kurabo Industries Ltd., Osaka, Japan) consisted of upper and lower chambers separated by a membrane with a pore size of 5 μm. Tregs (2.5–3.5 × 10^4^ cells) were placed in the upper chamber, and 4 ng/mL or 200 ng/mL of recombinant (r) IL-33 (PeproTech Inc., Rocky Hill, NJ, USA), rCCL17 (PeproTech Inc.), rCCL21 (PeproTech Inc.), or rCCL22 (R&D Systems) in culture medium were added to the lower chamber. After 2 h of incubation at 37 °C, the number of cells that migrated into the lower chamber was counted.

### 4.11. Statistical Analyses

All experiments were repeated at least three times. Data were expressed as mean values ± SEM (standard error of the mean). Differences between groups were examined for significance by one-way ANOVA with Sidak’s multiple-comparison test, two-way ANOVA with Tukey’s multiple-comparison test or the Kruskal–Wallis test followed by Dunn’s multiple comparisons test. Correlations between groups were examined by Spearman’s rank correlation test. *p*-values were analyzed using Prism 7 software (GraphPad Software, San Diego, CA, USA), with *p* < 0.05 indicating significance.

## Data Availability

All data are contained within the manuscript.

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
