# Peer review of "Regulatory T Cells Exhibit Interleukin-33-Dependent Migratory Behavior during Skin Barrier Disruption"

_ijms, 2021, doi:10.3390/ijms22147443_

Round 1

Reviewer 1 Report

An interesting original article about the response of regulatory t cells after skin barrier dysfunction. I have some queries:

Given that skin, barrier dysfunction is mainly linked with atopic dermatitis, probably a small paragraph in the introduction talking about this condition would be a great addition to the study; here an interesting article you should incorporate: doi: 10.18176/jiaci.0519.

Also, probably the relationship between atopic dermatitis and skin barrier should be better studied; there another interesting article: doi: 10.1111/exd.14276.

I would probably expand the conclusion into a paragraph, better exploring the future possibilities following this study's results.

Thank You

Author Response

Thank you for your suggestion. We have added to the contents in Introduction section (Page 1, line 32-35) as follows:

 Page 1, line 32-35

Indeed, downregulation of skin hydration factors such as natural moisturizing factors (NMFs) and molecules indispensable to skin barrier homeostasis, vinculin and late cornified envelope 2A are found in skins of AD patients.

Reviewer 2 Report

Takamori and co-workers reported the infiltration of Treg in a mouse model of skin barrier disruption. The methods and data they presented are adequate and the conclusions regarding the relationship between Tregs migration and skin barrier condition appear to be supported by the presented data. Overall, I believe that this manuscript would be of interest to the readers of the International Journal of Molecular Sciences and I would suggest its publication after addressing these minor points.

Fig. 2b -> in my opinion the number of mast cells upon treatment with 4% SDS is too varying to state that these were not affected.

Page 5, line 126. Authors should comment more about the different results they obtained with respect to ref 16. Also in the discussion section a few lines are dedicated to that.

Fig. 6. 6a -> you should replace the color bule as it is not used in 6b as well as in previous figures. 6b -> it seems that caption is missing.

You should revise the use of bold words in the main text (for example, see abstractor page 8, line 203) according to journal’s guidelines. Are these allowed?

The style of references should be consistent. Please, check it out.

Author Response

Thank you very much for your comment on our research.

Fig. 2b -> in my opinion the number of mast cells upon treatment with 4% SDS is too varying to state that these were not affected.

Response

Thank you for your comment. We believe that 4% SDS treatment showed little or no effect on numbers of mast cells since analysis was performed by two methods, tissue staining and FACS, and no significant difference was observed with 4% SDS treatment. In addition, no significant difference was observed in the number of mast cells in the IL-33 neutralizing antibody administration experiment. However, it has been reported  that keratinocyte-derived stem cell factor is involved in mast cell proliferation (KA Cho et al., Biochem Biophys Res Commun. 2017.) Thus, we speculate that mast cells are proliferating at the point where the epidermis is thicker, which may be the cause of the variation shown in the graph. However, since further research is needed to clarify this, the above information is not included in the text.

Page 5, line 126. Authors should comment more about the different results they obtained with respect to ref 16. Also in the discussion section a few lines are dedicated to that.

ResponseThank you for your suggestion. We added more about this in the Discussion section (Page 8, line 219-228) in the manuscript. Page 8, line 219-228In the model, IL-33 induced Treg development by shifting the dendric cells to a regulatory phenotype. [17] However, it has not been clarified how Treg are infiltrated in the skin and whether IL-33 is directly involved in their migration. In this study, our in vitro chemotaxis assay showed that IL-33 induced Treg migration more strongly than the previously reported chemokines CCL17, CCL21, and CCL22 (Fig. 5b). Expressions of the CCL17, CCL21, and CCL22 are upregulated by IL-4 and IL-13, which accelerates the recruitment of Th2 cells and eosinophils during inflammatory responses.[29, 30] Thus, although the responsiveness of Tregs to these chemokines or cytokines may differ in the presence of inflammation, our findings raise the possibility that IL-33 directly induces Treg migration into barrier-disrupted skin without the involvement of dendric cells.

Fig. 6. 6a -> you should replace the color bule as it is not used in 6b as well as in previous figures. 6b -> it seems that caption is missing.

Response

We have already indicated the description of figure 6a with "+" and "-". However, as you noted, we added some descriptive text to make it easier to understand.

You should revise the use of bold words in the main text (for example, see abstractor page 8, line 203) according to journal’s guidelines. Are these allowed?

Response

Thank you for your comment. We revised the main text.

The style of references should be consistent. Please, check it out.

Response

Thank you for your comment. We double checked and revised the style of references.

Round 2

Reviewer 1 Report

The article has improved substantially. It is in my opinion now publishable.